Pain without gain? A randomized crossover study on the impact of active and passive foam rolling on jump height and pain intensity

Heinke Lars 1
Javanmardi Sasha 1
Zemke Janis Alexander 1
Rappelt Ludwig 1 2
Freiwald Jürgen 1
Baumgart Christian 1
Niederer Daniel niederer@sport.uni-frankfurt.de 3
1 Department of Movement and Training Science, University of Wuppertal , Wuppertal , Germany
2 Department of Intervention Research in Exercise Training, German Sport University Cologne , Cologne , Germany
3 Institute of Occupational, Social and Environmental Medicine, Goethe University Frankfurt , Frankfurt am Main , Germany
Young Jesse
Electronic publication date: 2025 Jul 30
Publication date: 2025
Volume: 13
Electronic Location ID: e19747
Received 2025 Mar 13; Accepted 2025 Jun 23
Copyright: ©2025 Heinke et al.
Copyright year: 2025
Copyright holder: Heinke et al.
License: This is an open access article distributed under the terms of the Creative Commons Attribution License, which permits unrestricted use, distribution, reproduction and adaptation in any medium and for any purpose provided that it is properly attributed. For attribution, the original author(s), title, publication source (PeerJ) and either DOI or URL of the article must be cited.
License URL: https://creativecommons.org/licenses/by/4.0/

Keywords: Foam rolling, Myofascial release, Pain perception, Countermovement jump, Warm-up, Muscle

Funding: The authors received no funding for this work.

==============================
Background

Foam rolling has become increasingly popular for its proposed benefits on physical performance and recovery. This study investigated the effects of single bouts of active foam rolling and passive foam rolling on vertical jump height, perceived pain, and applied pressure during treatment.

Methods

Twenty physically active participants (10 males, 10 females) completed a randomized crossover design study, undergoing one active and one passive foam rolling session. Jumping performance was assessed via countermovement jump (CMJ) height at baseline, pre-treatment (PRE), and post-treatment (POST). Pain intensity was evaluated using a visual analog scale, while applied pressure was measured via force plates for active foam rolling and the weight applied to a custom device for passive foam rolling.

Results

The CMJ height post-treatment was reduced after both foam rolling treatments (p < 0.001, ωp2 = 0.29), with no significant interaction or condition effect observed. The applied pressure during active was significantly higher than during passive foam rolling for the thigh (p < 0.001, Hedges’ g = 1.14). In contrast, perceived pain was greater in passive than in active rolling (p = 0.002, Hedges’ g = 0.96). CMJ height improved following the initial warm-up (baseline to PRE, p = 0.014, 95%, Hedges’ g = −0.11).

Conclusion

This study highlights the biomechanical and psychological complexities of foam rolling, suggesting that both active and passive rolling may temporarily impair power performance. The observed jump height reduction could stem from decreased tissue stiffness, while the initial warm-up benefits reinforce the effectiveness of traditional warm-up protocols.

Introduction

Foam rolling (FR) has gained popularity as a self-massage technique, offering potential benefits in terms of increasing training efficiency and improving post-exercise recovery (Healey et al., 2014; Jones et al., 2015). Primarily categorized as a self-myofascial release technique, FR involves the application of direct pressure to muscles using tools such as foam rollers or massage bars/sticks (Peacock et al., 2014). Foam rolling positively impacts various aspects of physical performance and recovery, including performance, recovery, flexibility, pain, and the autonomic nervous system (Glänzel et al., 2023; Konrad et al., 2024; Peacock et al., 2015; Wiewelhove et al., 2019). Psychological, physiological, neurological, and biomechanical mechanisms are suggested to be mechanisms for these effects (Giovanelli et al., 2018; Glänzel et al., 2023; Macgregor et al., 2018; Wiewelhove et al., 2019).

However, the impact of FR on power type exercises remains uncertain, with conflicting findings in the existing literature: Specifically, the effects of different FR execution techniques, namely active foam rolling (AFR) and passive foam rolling (PFR), on power-specific performance outcomes are still unclear. On the one hand, studies on AFR combined with warm-up exercises have yielded mixed results. Richman, Tyo & Nicks (2019) demonstrated improved jump height when AFR was combined with light jogging and dynamic stretching, compared to a control group without AFR. Similarly, Peacock et al. (2014) found enhanced jump performance after AFR over a combination of dynamic warm-up techniques. Conversely, Phillips et al. (2021) and Sağıroğlu et al. (2017) reported decreased jump height after AFR, especially with longer AFR sessions. In these studies, the control group mimicked AFR positions without using a foam roller. However, Baumgart et al. (2019), Healey et al. (2014), Jones et al. (2015) and Smith, Pridgeon & Hall (2018) did not observe significant changes in jump height following AFR at all. On the other hand, passive foam rolling (PFR) has received very limited attention in research. A Study by Grabow et al. (2018), showed no impact of PFR using a constant-roll massage device on drop jump performance. Grabow et al. (2018) assessed the effects of PFR using three load levels determined by participants’ perceived discomfort levels on a 10-point visual analog scale (VAS).

Understanding the acute effects of different self-massage interventions is crucial for optimizing athletic performance and making informed decisions in both training and competition settings. Despite the growing use of foam rolling, there is still limited evidence comparing acute biomechanical and psychological impacts on performance. This study therefore aims to provide a comprehensive evaluation of the acute effects of single bouts of AFR and PFR on vertical jump height, with particular attention to the applied pressure on the tissue and the subjective perception of pain during treatment. The findings may offer valuable insights into the practical application and effectiveness of these widely used self-massage intervention. We hypothesized that (1) both FR strategies led to a comparable reduction in jumping height, that (2) PFR leads to higher mechanical forces than AFR, and that (3) AFR leads to comparable pain intensity perception when compared to PFR.

Materials & Methods

Participants

Twenty participants, comprising 10 males (average age: 26.3 ± 3.6 years, height: 177.0 ± 6.9 cm, weight: 78.5 ± 11.3 kg) and 10 females (average age: 24.2 ± 2.9 years, height: 166.4 ± 5.7 cm, weight: 64.2 ± 8.9 kg), were included in the study. No participant had to be excluded during study conduction, no one withdraw consent. No adverse effect occurred.

Measures

The primary outcome measure was the jumping height at a countermovement jump (CMJ). At each time point (Baseline, PRE, POST) during both visits, participants performed three maximal CMJs. Jumps were conducted before the warm-up (Baseline), after the warm-up but before foam rolling (PRE), and immediately after foam rolling (POST). To minimize upper body involvement and ensure consistency, participants kept their hands on their hips during all jumps. The vertical jump height was calculated using the impulse–momentum method (Linthorne, 2001). Ground reaction forces were recorded at a sampling rate of 1,000 Hz using two rigid force plates (Type 9287BA, Kistler, Winterthur, Switzerland). These force plates were connected to a personal computer running custom data acquisition software developed with LabView (version 9.0f2, National Instruments, Austin, TX, USA). The software converted the raw electrical signals (in volts) from the force plates into force data (in newtons) along the x-, y-, and z-axes. The processed force data were exported to a custom Excel file (.xlsx) for subsequent analysis.

The secondary outcomes were perceived pain intensity elicited by AFR and PFR and the applied pressure during the foam rolling treatments. Perceived pain intensity was assessed at POST using a visual analog scale (VAS) ranging from 0 to 10 cm. A score of 0 indicated no pain, and 10 indicated maximum perceived pain. In AFR the vertical force during the treatment was sampled at 1,000 Hz using a force plate (Type 9287BA, Kistler, Winterthur, Switzerland), and Noraxon MR 3 Version 3.16.84 software (Noraxon USA Inc., Scottsdale, AZ, USA) was used for data recording and processing. The center of pressure was analyzed to identify the repetitions and movement directions. After time normalization, a mean force curve was calculated in newtons and in percent of the body weight for each rolling repetition. The force curves were then averaged for all repetition. For PFR the applied pressure while rolling was individualized to 32.1 ± 1.5% of the participants body weight, following Baumgart et al. (2019).

Design and procedures

We adopted a randomized cross-over study design. Prior to measurements, participants received comprehensive information about the experimental procedures and provided full oral and written informed consent. The study’s adherence to the Declaration of Helsinki was confirmed by the independent University of Wuppertal ethics committee (MS/AE 211222).

Eligible participants were aged between 18 and 35 years and, according to self-report, met at least the World Health Organization’s minimum physical activity guidelines—corresponding to Tier 1 (Recreationally Active) in the participant classification framework proposed by McKay et al. (2022). To screen for contraindications related to the mechanical load on rolled structures, participants underwent a thorough assessment for the following specific exclusion criteria. This included checking for joint swelling, acute pain, cardiovascular or orthopedic issues, fibromyalgia, osteoporosis, rhabdomyolysis, rheumatic diseases, and the use of anticoagulant medications—all of which none of the participants reported meeting. All participants made two visits to the laboratory. A washout phase of 24 h was kept constant between visits. At each visit, three measurement were performed (Baseline, PRE, POST). Between PRE and POST, one of the two foam rolling protocols was adopted (Fig. 1). The sequence of foam rolling treatments, either AFR or PFR, within the framework of the study was randomized (full randomization) through the “Research Randomizer” program (Urbaniak & Plous, 2013). The allocation sequence was blinded to the investigator (electronically). Explicit instructions were given to participants, directing them to refrain from using a foam roller in their personal space during the study to prevent potential influences on subsequent measurements.

Figure 1 Study flowchart outlining the participant flow and key stages of the study procedure.

A 5-minute warm-up targeted general and jump-relevant lower limb muscles on a stationary bicycle ergometer. Participants’ individual load was determined using the age-dependent Fox formula (70% of max heart rate) (Fox & Naughton, 1972). In both FR treatments, a BLACKROLL® foam roller (30 cm length, 15 cm diameter) (BLACKROLL AG, Bottinghof, Switzerland) was used. Each time, a metronome set at 30 beats per minute coordinated rolling speed for both AFR and PFR, with each rolled path lasting two seconds. Both limbs were, one after the other, treated for 3 min, resulting in a total treatment length of 6 min per condition (AFR and PFR). Target muscles were the front thigh muscles of the limb and the gastrocnemius. The front thigh muscles were massaged from proximal to distal between the anterior superior iliac spine and the upper end of the kneecap and back along the same path for one minute. To target three portions of the front thigh (vastus medialis, rectus femoris, vastus lateralis) the treated limb was slightly rotated during treatment for 20 s each. The gastrocnemius muscle was treated from proximal to distal and back between the popliteal fossa and the myotendinous junction of the Achilles tendon for 30 s. The FR treatment always began with the left limb and then switched to the right. Two cycles of the described procedure were completed. For PFR treatment (see Fig. 2), a custom-made constant pressure rolling device was used, in which the foam roller was clamped. The apparatus‘s own weight was 11.3 kg. To apply the intended pressure to the tissue during PFR the device was loaded with additional weight as needed.

Figure 2 Illustration of PFR applied to the anterior thigh and calf muscles.

(A & B) Treatment of the anterior thigh muscles. The foam roller was applied from the anterior superior iliac spine to the upper edge of the patella and back for one minute. To target the vastus medialis, rectus femoris, and vastus lateralis, the limb was slightly rotated during treatment (20 s per portion). (C & D) Treatment of the gastrocnemius muscle. The foam roller was moved from the popliteal fossa to the myotendinous junction of the Achilles tendon and back for 30 s.

Statistical analysis

If not stated otherwise, all data are presented as means and standard deviations (SD). The data analysis procedures followed the approach previously described by Rappelt et al. (2024): Normal distribution was verified via the Shapiro–Wilk test (p ≥ 0.1) and investigation of residuals using Q–Q plots. Variance homogeneity was verified employing Levene-tests (p ≥ 0.1). No outliers (≥Q3 + 1.5 × interquartile range (IQR = Q3−Q1) or ≤ Q1−1.5  × IQR) were found in the raw data.

To assess the consistency of jumping performance at baseline between AFR and BFR, intraclass correlation coefficients (ICC) were calculated (two-way random model for consistency; ICC(2,1)). ICC evaluation was conducted in accordance to the recommendations of Koo & Li (2016). ICC ≥ 0,90 was rated as “excellent”, values between 0,90 to ≥ 0,75 as “good”, values between 0,75 to ≥ 0,50 as “moderate” and values < 0,50 as “poor”. To identify possible differences in jumping height between the two experimental conditions and the three time points, a linear mixed-effects model was fitted with fixed effects for condition (AFR vs. PFR) and time (Baseline, PRE, POST) and their interaction, and a random intercept for participant. Fixed effects were analyzed using F-tests (type III) with Satterthwaite approximations for the degrees of freedom. Model effect sizes are given as partial omega squared (ωp2), with 0.01 ≤ωp2 < 0.06, 0.06 ≤ωp2 < 0.14, ωp2 ≥ 0.14 indicating small, moderate, and large effects, respectively (Cohen, 1988). Subsequently, in case of a statistically significant interaction effect or main effect of time, Tukey post-hoc tests to adjust for multiple testing were computed. For pairwise effect size comparison, Hedges’g was calculated to interpret effect sizes as small (0.2 ≤ g < 0.5), medium (0.5 ≤g < 0.8) or large (g ≥ 0.8) (Hedges & Olkin, 1985). Moreover, to identify possible differences in applied pressure and perceived pain, dependent, two-sided t-tests were calculated. Again, Hedges’g was calculated to interpret effect sizes. All statistical analyses and visualizations were conducted using R (version 4.2.0) in RStudio (version 2023.06.1+524), as outlined by Rappelt et al. (2024). Specifically, the ICC analysis was conducted using the ICC()-function from the psych package (Revelle, 2024). The package rstatix (Kassambara, 2023) was used for analysis of outliers and performing Shapiro–Wilk tests. For Levene-tests the car package (Fox et al., 2023) was used. Mixed modelling was performed using the lmerTest package (Kuznetsova, Brockhoff & Christensen, 2017) and the anova()-wrapper of stats (part of base R) was used to provide ANOVA tables. For computing t-tests, again the stats package of base R was used. Effect size estimation was performed using the effectsize package (Ben-Shachar, Lüdecke & Makowski, 2020). Post-hoc testing (Tukey) was performed via the emmeans package (Lenth et al., 2022). For all statistical analyses, the level of statistical significance was set at α = 5%.

Results

Jumping performance

The calculated ICC(2,1) point estimate for jumping height at baseline between the two conditions can be considered as excellent (ICC(2,1) = 0.969, 95% confidence interval (CI) [0.925–0.988]). The CMJ neither exhibited a statistically significant interaction effect (F(2, 95) = 1.64, p = 0.287, ωp2 < 0.001 (small effect size)) nor a statistically significant main effect for condition (F(1, 95) = 2.72, p = 0.103, ωp2 = 0.02 (small effect size)). However, a statistically significant effect for time was observed (F(2, 95) = 20.77, p < 0.001, ωp2 = 0.29 (large effect size)). Subsequent post-hoc analysis revealed statistically significant differences between Baseline and PRE (t(95) = −0.286, p = 0.014, 95% CI [−0.122, −1.339], Hedges’g = −0.11 (trivial effect size)), Baseline and POST (t(95) = 3.57, p = 0.002, 95% CI [0.304–1.520], Hedges’g = 0.15 (trivial effect size)) as well as PRE and POST (t(95) = −6.43, p < 0.001, 95% CI [−1.035, −2.251], Hedges’g = 0.28 (small effect size)). Consequently, CMJ jumping height slightly increased from Baseline (AFR: 0.265 ± 0.064 m, 95% CI [0.234 m–0.295 m], PFR: 0.264 ± 0.063 m, 95% CI [0.234 m–0.293 m]) to PRE (AFR: 0.272 ± 0.061 m, 95% CI [0.244 m–0.301 m], PFR: 0.271 ± 0.059 m, 95% CI [0.243 m–0.298 m]), regardless of the condition and subsequently decreased towards POST (AFR: 0.259 ± 0.057 m, 95% CI [0.233 m–0.286 m], PFR: 0.251 ± 0.055 m, 95% CI [0.225 m–0.277 m]) (Fig. 3).

Perceived pain and applied pressure

Differences were found between conditions in terms of perceived pain (AFR: 3.90 ± 2.36, 95% CI [2.79–4.99] vs. PFR: 6.06 ± 2.06, 95% CI [5.09–7.02], t(19) = 3.58, p = 0.002, 95% CI [−1.20, 1.90], Hedges’g = 0.96 (large effect)) and mean applied pressure to the thigh (AFR: 291.0 ± 74.1 N, 95% CI [256.3 N–325.7 N] vs. PFR: 223.7 ± 53.0 N, 95% CI [207.3 N–240.1 N], t(19) = 5.58, p < 0.001, 95% CI [42.10–92.59], Hedges’g = 1.14 (large effect)). However, mean applied pressure to the calf did not differ between conditions (AFR: 226.8 ± 43.5 N, 95% CI [206.4 N–247.2 N] vs. PFR: 223.7 ± 53.0 N, 95% CI [207.3 N–240.1 N], t(19) = 0.59, p = 0.561, 95% CI [−7.86, 14.04], Hedges’g = 0.08 (trivial effect)) (Table 1).

Figure 3 Boxplot (Q1 to Q3, including median), and whiskers (showing minimum and maximum values) of countermovement jump results for the active (AFR; blue) and passive (PFR; grey) foam rolling condition.

Individual values are presented as dots which are connected for every participant.

Table 1 Comparison of applied pressure related to body weight and the perceived pain (VAS) between the passive and active foam rolling condition.

Parameter	Passive foam rolling	Active foam rolling	Mean difference
(95% CI)	Standard error	t-statistic (df)	p-value	Hedges’ g
(95% CI)	
Mean pressure thigh
(% bw)	32.1 ± 1.5	41.6 ± 7.0	9.5
[6.3, 12.7]	1.5	t (19) = 6.29	<0.001	1.85
[1.10, 2.57]	
Mean pressure calf
(% bw)	32.1 ± 1.5	32.4 ± 23.0	0.4
[−1.2, 1.9]	0.74	t (19) = 0.65	0.527	0.15
[−0.46, 0.76]	
Perceived pain (VAS)	6.06 ± 2.06	3.90 ± 2.36	−2.2
[−3.4, −0.9]	0.6	t (19) = −3.58	0.002	0.96
[0.30, 1.61]	
Notes.

% bw percentage of body weight

df degrees of freedom

95% CI 95%-Confidence Interval

Discussion

This study assessed the acute impact of single bouts of active foam rolling (AFR) and passive foam rolling (PFR) on vertical jump height in the countermovement jump (CMJ) and (2) explored the relationship between participants’ pain perception and the applied pressure during AFR and PFR. Our main findings were: (1) both AFR and PFR led to a reduction in jumping height, and (2) Comparing the applied pressure in AFR and PFR, significantly higher mechanical forces in AFR does not lead to increased pain perception compared to PFR. The hypothesis (1) can, thus, be supported; whereas the hypothesis (2) and (3) cannot be supported. Analysis of consistency in jump height for Baseline indicated excellent reproducibility. Consequently, the risk of potential between day carry over effects due to habituation can be treated as minimal. Following both AFR and PFR sessions, a reduction in vertical jump height during CMJs occurred. These findings are consistent with those studies highlighting the negative effects of AFR on jumping performance (Phillips et al., 2021; Sağıroğlu et al., 2017). However, variations in foam rolling protocols and post-treatment measurement timing complicate direct comparisons. For instance, Phillips et al. (2021) reported a significant decrease in jump height five minutes after foam rolling, whereas no adverse effects were observed when measurements were taken one minute post-treatment. Similarly, in the study by Sağıroğlu et al. (2017), participants used a distinct foam rolling cadence—five rolls per 30 s with maximum pressure—over a total duration of eight minutes (Guest et al., 2021).

The observed negative effect of PFR in our data contrasts with the findings of Grabow et al. (2018), who evaluated the effects of PFR on power-type exercises using a single-leg drop jump and found no such detrimental effects. The decline in jump height observed under both treatment conditions in our study may be linked to the effects of foam rolling on tissue stiffness. A meta-analysis by Glänzel et al. (2023) qualitatively suggests that foam rolling reduces fascial and muscle stiffness, which could impact power-based exercises. However, the authors found no conclusive evidence that it alters myofascial tissue stiffness. Additionally, Habscheid, Szikszay & Luedtke (2024) highlight in their review that the pressure applied during foam rolling may be too low to affect fascial tissue, leaving the underlying mechanisms behind the observed performance decline unclear. Additionally, the effects of single bouts of foam rolling on the rate of force development remain unclear due to the limited amount of available research, which hampers the ability to conduct a comprehensive quantitative analysis. The only relevant study, conducted by MacDonald et al. (2013), examined the impact of two 1-minute foam rolling treatments and reported no significant changes in the rate of force development. One may speculate that the overall higher dosage of foam rolling used in our study may have contributed to a modulation of muscle and tissue stiffness, resulting in the more pronounced negative effects on performance. This hypothesis warrants further investigation, particularly considering the potential impact of non-contracted musculature during PFR, which may allow for greater alterations in tissue stiffness.

In contrast to the decline in jump height observed after AFR and PFR, CMJ height increased immediately following the warm-up. As demonstrated by Tsurubami et al. (2020), performance benefits can be achieved through a moderate-intensity warm-up. Consequently, a traditional non-specific warm-up that targets jump-relevant limb muscles appears to be more effective in enhancing vertical jump performance (Fletcher, 2013).

We evaluated the biomechanical load applied to the thigh and calf during FR treatments by measuring vertical ground reaction forces during AFR and quantifying the weight added to the custom-made constant pressure rolling device during PFR. Our results showed that the mean forces applied to the thigh were significantly higher during AFR compared to PFR. This difference was observed in both absolute terms and relative to participants’ body weight. Additionally, AFR demonstrated substantially higher peak forces, with maxima reaching up to 55.3% of participants’ body weight. At the calf, however, no significant differences were observed in biomechanical load between AFR and PFR. As Curran, Fiore & Crisco (2008) noted, the pressure applied during FR treatments can be twice as high as that used in occlusion studies. This indicates that both AFR and PFR in our study induce considerable mechanical compression on underlying tissues, potentially impacting connective tissue, nerves, vessels, and bones (Freiwald et al., 2016). Future studies should explore these effects further, especially given the inverse relationship we observed between biomechanical loading and pain perception. Interestingly, despite the significantly higher pressure applied during AFR—reaching up to 55.3% of participants’ body weight at the thigh—it did not result in greater pain perception compared to PFR. The high VAS scores for pain perception following PFR suggest that participants may have underestimated the pain experienced during AFR. This phenomenon might be explained by an increase in the pressure pain threshold due to underlying muscle contractions during AFR. As Habscheid, Szikszay & Luedtke (2024) demonstrated foam rolling is associated with changes in pressure pain threshold. Mense (2008) proposed that free nerve endings of nociceptors can become sensitized and activated by strong mechanical stimuli, such as overloading. Supporting this, Kosek & Ekholm (1995) demonstrated a significant increase in the pressure pain threshold during a 5-minute contraction at 21% MVC of the M. quadriceps femoris. This suggests that neural input from both cutaneous and deeper tissues interacts with nociceptive activity, potentially modulating pain perception during muscle contraction.

While our study provides valuable insights with practical relevance, several limitations of the study design must be acknowledged. First, we focused exclusively on the acute effects of single bouts of FR on jumping performance, leaving the chronic adaptations to repeated AFR and PFR usage unexplored. Second, our study lacks an a priori sample size estimation. Third, the participant pool consisted of healthy, physically active young adults, which may limit the generalizability of our findings to other populations, such as older individuals, those with musculoskeletal conditions, or athletes accustomed to varying foam rolling techniques.

Conclusions

This study showed that: (1) both AFR (−4.8% ± 4%, 95% CI [−6.6%, −2.9%]) and PFR (−7.1% ± 4.2%, 95% CI [−9.3%, −5.3%]) led to a reduction in jumping height, and (2) Comparing the applied pressure in AFR and PFR, significantly higher mechanical forces in AFR does not lead to increased pain perception compared to PFR. Hypothesis (1) was supported by the results, whereas hypotheses (2) and (3) were not supported. The observed reduction in jumping performance following both AFR and PFR raises questions about the usefulness of foam rolling as a pre-exercise routine, especially given that both treatments caused substantial pain without any performance gains. Concerns regarding potential adverse effects on tissues should be carefully considered, particularly with AFR, which exhibited significantly higher pressure. Our findings emphasize the importance of safety considerations and further research to elucidate the mechanisms and long-term effects of different FR modalities.

Supplemental Information

Supplemental Information 1 Raw data

The authors thank Phillip Zivic for his support in data acquisition.

Additional Information and Declarations

Competing Interests

Author Contributions

Human Ethics

Data Availability

The authors declare there are no competing interests.

Lars Heinke conceived and designed the experiments, performed the experiments, analyzed the data, prepared figures and/or tables, authored or reviewed drafts of the article, and approved the final draft.

Sasha Javanmardi conceived and designed the experiments, performed the experiments, analyzed the data, authored or reviewed drafts of the article, and approved the final draft.

Janis Alexander Zemke conceived and designed the experiments, performed the experiments, analyzed the data, authored or reviewed drafts of the article, and approved the final draft.

Ludwig Rappelt conceived and designed the experiments, performed the experiments, analyzed the data, prepared figures and/or tables, authored or reviewed drafts of the article, and approved the final draft.

Jürgen Freiwald conceived and designed the experiments, authored or reviewed drafts of the article, and approved the final draft.

Christian Baumgart conceived and designed the experiments, performed the experiments, analyzed the data, authored or reviewed drafts of the article, and approved the final draft.

Daniel Niederer conceived and designed the experiments, analyzed the data, authored or reviewed drafts of the article, and approved the final draft.

The following information was supplied relating to ethical approvals (i.e., approving body and any reference numbers):

Ethical approval was granted by the independent University of Wuppertal ethics committee.

The following information was supplied regarding data availability:

The raw data are available in the Supplementary file: Data S1.

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
