# Peer review of "Pain without gain? A randomized crossover study on the impact of active and passive foam rolling on jump height and pain intensity"

_PeerJ, doi:10.7717/peerj.19747_

## Round 0.1 · original submission · Major Revisions

Three peer reviewers have reviewed this manuscript. Whereas Reviewers 1 and 2 found only minor issues, primarily suggesting ways to improve presentation clarity, Reviewer 3 had more major concerns. Though Reviewer 3 recommends rejection, I feel that all of the issues identified are addressable with some additional discussion of study rationale and methodological decisions. I am therefore recommending major revision.

Reviewer 1 ·

Basic reporting

Although the paper is well-written, several points require minor revisions, which are indicated in the main text.

Experimental design

-

Validity of the findings

-

Additional comments

Good luck!

Annotated reviews are not available for download in order to protect the identity of reviewers who chose to remain anonymous.

Reviewer 2 ·

Basic reporting

1. The authors indicated, "There is still a gap in understanding the biomechanical aspects and acute effects on performance as well as the psychological factors associated with potential performance alterations." Filling a gap should not be the purpose of the study. If a gap is not important, there is no need to fill the gap. I would like to suggest the importance of the study.
2. The authors indicated "Intraclass Correlation Coefficients (ICC) were calculated (two-way random model for consistency; ICC(2,1)). The ICCs were rated as excellent (0.9 to 1), good (0.74 to 0.9), moderate (0.4 to 0.73), and poor (0 to 0.39)." Please show your reference.
3. The authors indicated "Model effect sizes are given as partial omega squared (Ëp2), with >0.01, >0.06, >0.14 indicating small, moderate, and large effects, respectively (Cohen 1988)." That range should be mutually exclusive. for example, if a wp2 is 0.5, it is > 0.01, > 0.06, and > 0.14 based on the criterion. However, it cannot be in small, moderate, and large effects.
4. Figure 1 does not look good. Please replot it.

Experimental design

1. Please show what APR and PRF are. It is hard to understand active foam rolling.
2. The authors indicated "To identify possible differences in jumping height between the two experimental conditions and the three time points, a 2 (condition: AFR vs. PFR) ×)3 (time: Baseline, PRE & POST) design was used. Condition and time were modelled as fixed effects, while participants were included as a random effect blocking factor." Is this just 2-way repeated measure ANOVA?
3. The authors indicated "To assess the consistency of jumping performance at baseline between AFR and BFR, Intraclass Correlation Coefficients (ICC) were calculated (two-way random model for consistency;" Please explain why there is a need to use ICC. The main purpose is to compare performance between conditions, and the can be done by using ANOVA.

Validity of the findings

1. Hypotheses (1) and (2) were supported rather than accepted. A hypothesis can never be accepted.
2. The authors indicated "This study showed that: (1) both AFR and PFR led to a reduction in jumping height," "our" is not necessary. Also, please report a reduction of jumping height in %.

Additional comments

1. It definitely needs a figure showing the data collection.
2. It also needs a figure to show what the rolling foam looks like.

Reviewer 3 ·

Basic reporting

This study investigated the effects of single bouts of active and passive foam rolling (FR) on vertical jump height, perceived pain, and applied pressure during FR. While the experiment appears to have been conducted under well-controlled conditions, the most critical issue is the insufficient description of the active and passive FR protocols. As a result, I was unable to clearly understand the treatments or how they differed, which made it difficult to fully grasp the study's content.

1. The title is catchy; however, since the conclusions do not discuss either "pain" or "gain," I recommend revising it to better reflect the actual findings.
2. Please cite relevant literature on the effects of FR on subsequent performance. While the introduction mentions the combination with warm-up and differences in FR duration, it is unclear how those findings relate to the present study. It should be clearly stated how previous studies informed this work, specifically, whether the current protocol is based on those found effective in prior research.
3. Since I could not understand what distinguishes active from passive FR, the rationale for conducting this experiment was unclear. Moreover, although pain is referenced in the title, it is not addressed in the Introduction.
4. I found it difficult to visualize the experimental setups for AFR and PFR. Please include a figure or photograph to illustrate each condition.
5. Lines 107–108: The analysis of the center of pressure (COP) is mentioned, but corresponding results are not reported.
6. Lines 110–111: What pressure level was the rolling set to to achieve the reported average of 32.1%?
7. Table 1: Please also include the pain outcomes here.
8. Lines 114–119: This content would be more appropriately placed in the Participants section.
9. Line 225: The citation "(Guest et al., 2021)" should be placed before the period.

Experimental design

The study appears to be well-controlled and conducted to a high standard. However, the methods used for the active and passive foam rolling interventions are not clearly described. Detailed information regarding how each treatment was administered is necessary to ensure reproducibility and to allow readers to properly interpret the experimental conditions.

Validity of the findings

The results are clearly presented and appear valid in isolation; however, the Introduction lacks a clear rationale and specific hypotheses, making it difficult to contextualize the findings. Additionally, due to the insufficient description of the intervention protocols, I was unable to fully understand the implications of the results. The authors should clarify what is already known from previous studies and clearly state the novel contribution of this study in the Introduction. This would strengthen the logical flow from background to interpretation.

---

## Round 0.2 · accepted · Accept

I commend the authors on an excellent job responding to all the the reviewer comments from the first version of the manuscript. I have heard back from two of the three previous reviewers, and both recommend the manuscript be accepted as is. I concur with this recommendation.

Reviewer 1 ·

Basic reporting

Since my previous evaluation, I have decided that the paper can be accepted after minor revision. The corrections made by the author are satisfactory. The paper is suitable for publication.
Regards,

Experimental design

Since my previous evaluation, I have decided that the paper can be accepted after minor revision. The corrections made by the author are satisfactory. The paper is suitable for publication.
Regards,

Validity of the findings

Since my previous evaluation, I have decided that the paper can be accepted after minor revision. The corrections made by the author are satisfactory. The paper is suitable for publication.
Regards,

Additional comments

Hı,
Thank you for your cooperation. Regards,

Reviewer 2 ·

Basic reporting

the authors have addressed my concerns in the previous version. I am good with the revision.

Experimental design

NA

Validity of the findings

My only major concern in he previous version is "accepting" the hypothesis. It has been corrected.

Additional comments

NA